# Geometry and evolution of the ecological niche in plant-associated microbes

Thomas M. Chaloner[1], Sarah J. Gurr [1,2] & Daniel P. Bebber [1✉]

The ecological niche can be thought of as a volume in multidimensional space, where each dimension describes an abiotic condition or biotic resource required by a species. The shape, size, and evolution of this volume strongly determine interactions among species and influence their current and potential geographical distributions, but the geometry of niches is poorly understood. Here, we analyse temperature response functions and host plant ranges for hundreds of potentially destructive plant-associated fungi and oomycetes. We demonstrate that niche specialization is uncorrelated on abiotic (i.e. temperature response) and biotic (i.e. host range) axes, that host interactions restrict fundamental niche breadth to form the realized niche, and that both abiotic and biotic niches show limited phylogenetic constraint. The ecological terms 'generalist' and 'specialist' therefore do not apply to these microbes, as specialization evolves independently on different niche axes. This adaptability makes plant pathogens a formidable threat to agriculture and forestry.

[1] Department of Biosciences, University of Exeter, Exeter EX4 4QJ, UK. [2] Department of Biosciences, Utrecht University, Paduallaan 8, Netherlands. ✉email: d.bebber@exeter.ac.uk

The niche is a fundamental concept in ecology and evolution, describing the range of conditions under which an organism can survive and reproduce[1]. Hutchinson's model of the niche as a volume in multidimensional space[2], where each dimension represents an environmental condition or resource requirements affecting a species, has proven a powerful tool for understanding competition, trait evolution, ecological specialization, community assembly rules, and the distributions of species on Earth[3,4]. In the era of anthropogenic habitat modification, climate change, and invasive species, modelling the ecological niche is key to predicting and mitigating the impacts of human activities on the biosphere[5]. Understanding the emergence of crop pathogens is of particular concern for global food security[6].

Niche theory differentiates between abiotic conditions, such as temperature or soil pH, and biotic resources, like host or prey availability. Abiotic conditions are unaffected by the species while resources can be depleted and competed over with other species, resulting in exclusion of the inferior competitor[3,7]. Biotic interactions thereby modify our expectations of where a species could exist in nature, reducing and altering the shape of the realized niche in comparison with the fundamental niche[1]. However, details of the geometry of the niche remain unresolved[4], such as the shape of the response of metabolic rates to temperature[8,9], the influence of biotic interactions on the abiotic niche[10,11], and the ability of species to specialize independently on abiotic conditions and biotic resources[12].

Here, we analyse temperature response functions and host ranges of hundreds of plant-associated fungi and oomycetes to understand the shape and size of an abiotic niche axis, and test whether these abiotic and biotic niches are correlated or independent. For plant pathogens, flexible and independent evolution on different niche axes would facilitate emergence and exacerbate the threat to crop production. Most biogeographical and niche

modelling studies have been conducted on plants, vertebrates and insects for which distributional data are available at high spatial resolution, and for which important biotic interactions, such as host or prey species, are known[11,13]. Much less is known about the niche dimensions of microbes, which are highly diverse and key to ecosystem function[14].

## Results

**Temperature response functions**. We collated and analysed temperature responses, specifically the minimum ($T_{min}$), optimum ($T_{opt}$) and maximum ($T_{max}$) temperatures that comprise the 'cardinal temperatures', of various biological processes for 695 plant-associated microbes (631 fungi and 64 oomycetes) cited in ref. [15] (Fig. 1). Cardinal temperatures can be used to derive temperature response functions, or thermal performance curves[9], using mathematical forms such as the beta function[16]. The biological processes for which cardinal temperatures have been measured vary in their degree of host interaction. Experimental measurements for rates of growth in culture (GC) and often spore germination (SG) occur under axenic conditions, while infection (IN) and disease development (DD) occur as interactions with the host plant. Fruiting body formation, or fructification (FR) and sporulation (SP) may or may not be measured in planta depending on experimental conditions. Variation in host interaction among biological processes allows the effect of biotic interactions on the temperature niche to be quantified.

We found substantial overlap in the distributions of cardinal temperatures between fungi and oomycetes for all processes (Fig. 1, Supplementary Table 1). GC and SG had somewhat lower $T_{min}$ and higher $T_{max}$ (and hence wider $T_{range}$) than other processes (Supplementary Table 1). Rates increase with temperature to $T_{opt}$ then decline to $T_{max}$, following thermodynamic

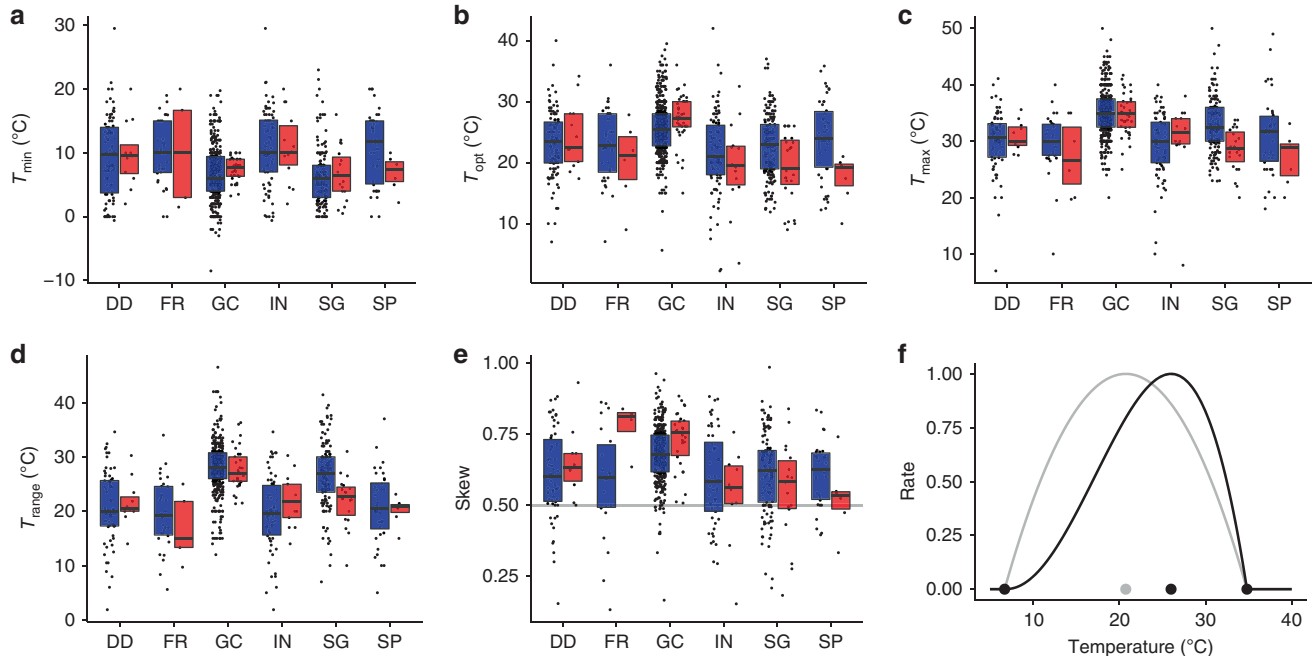

**Fig. 1 Temperature responses of plant-associated fungi (blue) and oomycetes (red) life history processes. a** Minimum temperature ($T_{min}$), **b** optimum temperature ($T_{opt}$), **c** maximum temperature ($T_{max}$), **d** temperature range ($T_{max}$–$T_{min}$). **e** Skew, where values >0.5 indicate the $T_{opt}$ is closer to $T_{max}$ than $T_{min}$. The grey line indicates where $T_{opt}$ lies half-way between $T_{min}$ and $T_{max}$. Processes are disease development (DD), fruitification (FR), growth in culture (GC), infection (IN), spore germination (SG), and sporulation (SP). **a–e** Boxplot boundaries reflect the inter-quartile range, the horizontal bar is the median. Summary statistics including sample sizes are reported in Supplementary Table 1. **f** Illustration of skew for a temperature response function. The black points show cardinal temperatures, with midpoint between $T_{min}$ and $T_{max}$ in grey. Unskewed response in grey, skewed response in black.

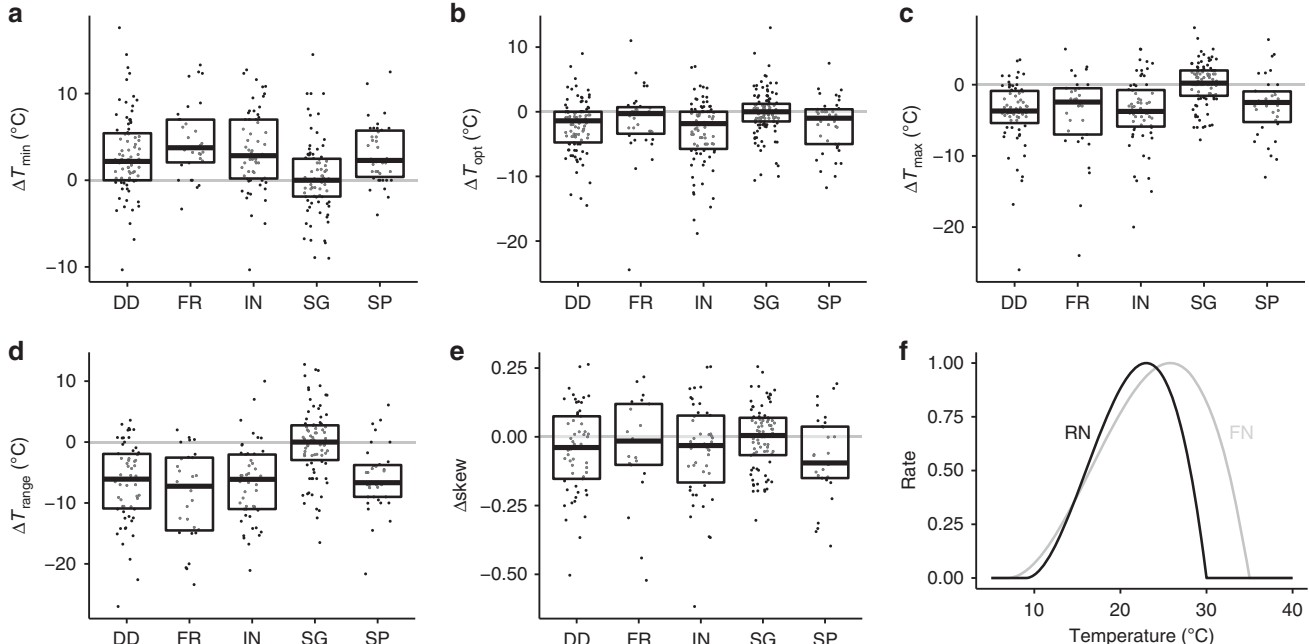

**Fig. 2 Temperature response differences to growth in culture. a** $T_{min}$, **b** $T_{opt}$, **c** $T_{max}$, **d** $T_{range}$ and **e** Skew. Processes are disease development (DD), fruitification (FR), infection (IN), spore germination (SG) and sporulation (SP). **a–e** Boxplot boundaries reflect the inter-quartile range, the horizontal bar is the median. Summary statistics including sample sizes are reported in Supplementary Table 2. **f** Illustration of temperature response (beta function) for the fundamental niche (grey, represented by growth in culture) compared with the realized niche (black, represented by disease development), where RN has a narrower $T_{range}$ and lower $T_{opt}$ than FN.

expectations[17]. We defined asymmetry, or skewness, of the temperature response function as the degree to which $T_{opt}$ is closer to $T_{max}$ (skew > 0.5) or $T_{min}$ (skew < 0.5). For nearly all processes in both fungi and oomycetes, $T_{opt}$ was closer to $T_{max}$ than $T_{min}$, but most strongly for GC (Fig. 1e). This suggests a difference in the shape of the temperature response for growth in axenic culture (GC) than for processes that involve interaction with the host plant or occur without nutrient media.

Within species, GC and SG tended to have similar cardinal temperatures (Fig. 2, Supplementary Table 2). GC and SG had lower $T_{min}$ than the other biological processes, higher $T_{opt}$, higher $T_{max}$, and a wider $T_{range}$ (Fig. 2). However, only SP had substantially lower skew compared to GC (Fig. 2e). $T_{opt}$ values were largely correlated across biological processes (Pearson correlation >0.6 for most processes, Supplementary Table 3), but $T_{range}$ values were weakly correlated (Supplementary Table 4), other than between DD and IN (Pearson correlation 0.91, 95% confidence interval 0.85–0.95, df = 44, t = 14.82, $p = 10^{-18}$). This likely resulted from the same cardinal temperature data being independently recorded as both IN and DD (see "Methods" section). Species are therefore warm or cold-adapted across biological processes, but there is less evidence that temperature niche breadth is correlated across processes. In summary, $T_{opt}$ and $T_{range}$ of the temperature response function were significantly greater for GC and SG than for other processes (Fig. 2e). This phenomenon has been detected in wild temperature-adapted strains of a fungal species[18]. The narrower temperature responses for in planta processes compared with in vitro processes could demonstrate the modification of the fundamental niche by biotic interactions to give the realized niche[1]. GC occurs under controlled axenic conditions with optimal nutrient availability, and in the absence of competition or other biotic interactions. Processes relating to disease in planta occur in the presence of plant host defences or stress responses and under nutrient restriction compared with processes occurring in culture media,

and so can be considered sub-optimal for the pathogen. These suboptimal resource conditions appear to restrict temperature niche breadth, reducing $T_{opt}$ by reducing the relative growth rate at higher temperatures (Fig. 2e). The left-skew of temperature response functions means that reduction in relative rates at high temperatures is much larger than at low temperatures (Fig. 2f).

**Niche co-specialization.** Species that occupy relatively large volumes of niche space are commonly described as generalists, while those with narrow tolerances are termed specialists[12]. There is little empirical understanding or theoretical consideration of the correlation between niche breadth on different niche axes, i.e. is the n-dimensional hyper-volume an n-sphere or a hyper-ellipsoid? We found no evidence for correlation between phylogenetic diversity of known host plants and $T_{range}$, indicating that specialization can occur independently for biotic resources and abiotic conditions (Fig. 3, Supplementary Table 5). The terms specialist and generalist therefore cannot be applied as holistic descriptions of fungal or oomycete species' ecology, but it is difficult to speculate on the selection pressures that would lead to differential specialization on abiotic and biotic niches.

**Phylogenetic signal in temperature response.** We investigated the evolution of temperature response and host range in the oomycete genus *Phytophthora* to determine which, if either, niche axis is under stronger phylogenetic constraint and thus more likely to control geographical distributions. We studied *Phytophthora* because this was the only multi-species genus in the dataset for which well-resolved molecular phylogenies and host range data were available[19]. $T_{opt}$ and $T_{max}$ of GC for 101 *Phytophthora* species extracted from ref. [20] were used for this analysis. We found a small but statistically significant phylogenetic signal in $T_{opt}$ for *Phytophthora* species (Bayesian phylogeny: Blomberg's $K = 0.226$, $p = 0.04$; maximum-likelihood phylogeny:

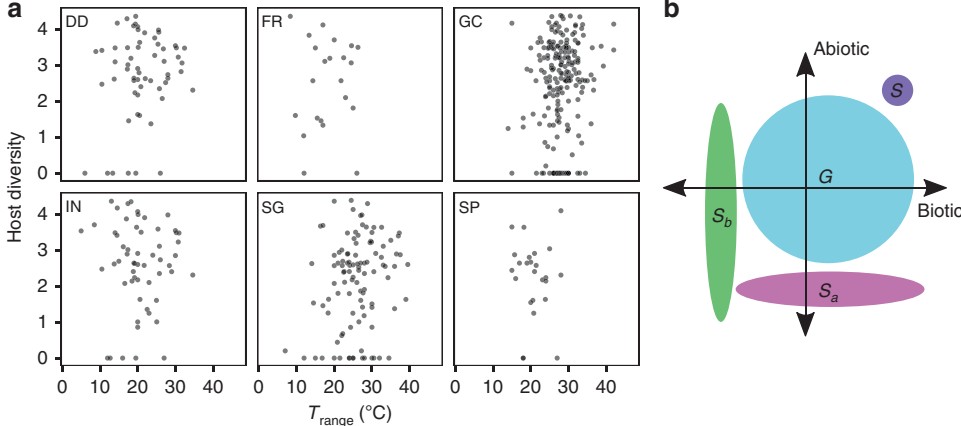

**Fig. 3 Biotic vs. abiotic niche breadth. a** Biotic niche breadth is represented by log-transformed host phylogenetic diversity calculated from the processed host phylogeny, and abiotic niche breadth by $T_{range}$. Panels show the different biological processes, disease development (DD, $N = 59$), fruitification (FR, $N = 22$), growth in culture (GC, $N = 209$), infection (IN, $N = 57$), spore germination (SG, $N = 105$) and sporulation (SP, $N = 25$). **b** Schematic of niche breadth on biotic and abiotic axes for a true specialist ($S$, purple), true generalist ($G$, blue), biotic specialist with wide climatic tolerances ($S_b$, green) and abiotic specialist with broad host range ($S_a$, magenta).

Blomberg's $K = 0.228$, $p = 0.001$; maximum parsimony phylogeny: Blomberg's $K = 0.700$, $p = 0.03$), and for $T_{max}$ (Bayesian phylogeny: Blomberg's $K = 0.804$, $p = 0.001$; maximum-likelihood phylogeny: Blomberg's $K = 0.259$, $p = 0.001$; maximum parsimony phylogeny: Blomberg's $K = 1.67$, $p = 0.001$). While closely related *Phytophthora* species had more similar thermal physiology than random pairs, in all but one case differences were greater than that expected under a Brownian motion evolutionary model[21,22]. This could not be explained by known geographical locations of species (Supplementary Fig. 1). Apparent latitudinal range shifts of plant pathogens in response to global warming[6] suggest niche conservatism in thermal physiology, i.e. migration is the dominant response of populations to changing climates rather than adaptation to new climates in situ[23]. There is some evidence for thermal adaptation in fungal pathogens[24] and within *Phytophthora*[25]. Our analysis suggests limited phylogenetic constraint in temperature niche evolution in *Phytophthora*, particularly for $T_{opt}$, though we acknowledge that a significant but small phylogenetic signal ($K < 1$) can arise from several different evolutionary models[26].

**Pathogen and host co-phylogenetic association.** In the evolution of host range, closely related plant species share pathogens[27] but the degree to which closely related pathogens share plant hosts is unclear. *Formae speciales* of powdery mildews, for example, are specialized upon, but not absolutely restricted to, particular plant hosts[28]. Evidence for different types of co-phylogenetic dynamics in plant–fungus symbioses, ranging from close congruence indicating co-divergence to incongruence, indicate long-range host switching[29]. Host jumps (acquisition of a host phylogenetically distant from current hosts) and transitions from specialist to generalist or vice versa are known in plant pathogens[30], suggesting that host range could be more evolutionarily labile than temperature physiology. We found a small but statistically significant co-phylogenetic association between the topologies of three *Phytophthora* phylogenies and the phylogeny of their plant hosts (439 species-level pathogen–host interaction records; Bayesian phylogeny: best-fit Procrustean super-imposition ($m^2_{XY}$) $= 0.939$, $p < 0.001$, median network randomization $=$ $0.976$ (IQR $0.975$–$0.978$); maximum-likelihood phylogeny: $m^2_{XY} = 0.939$, $p < 0.001$, median network randomization $= 0.976$ ($0.975$–$0.978$); maximum parsimony phylogeny: $m^2_{XY} = 0.935$, $p < 0.001$, median network randomization $= 0.974$ ($0.973$–$0.976$))

(Supplementary Fig. 2). Wide-ranging host jumps are known in *Phytophthora*, for example clade 1c (*P. infestans*, *P. ipomoeae*, *P. mirabilis*, and *P. phaseoli*) evolved through an ancestral major host jump, followed by adaptive specialization to one of four plant families, respectively[31]. *P. infestans* has been recorded on 22 host species, 20 in the Solanaceae and 2 in the sister-family Convolvulaceae, while *P. cactorum* infects the gymnosperm *Abies balsamea* as well as diverse angiosperm species.

## Discussion
Our analysis of the cardinal temperatures of plant-associated fungi and oomycetes shows that abiotic fundamental niches, as measured by temperature response functions, are wider than their corresponding realized niches. We show that microbial specialization can occur independently in abiotic (temperature response) and biotic (host range) niche axes, suggesting that the terms "specialist" and "generalist" should be used cautiously when describing the ecology of microbial species. Figuratively speaking, generalists have been characterized as "Jack of all trades, master of none". Our analyses suggest that "Jack of some trades, master of others" is more appropriate for species which have specialized on some niche axes. Finally, we show that both the thermal niche and host ranges are evolutionarily labile within genus *Phytophthora*, but retain weak phylogenetic signal.

Our analyses included only two niche axes, one abiotic and one biotic. Though this limits the generalizability of our conclusions, the axes we chose are amongst the most commonly studied, and are of fundamental ecological importance in determining species distributions and performance. Several analyses have investigated patterns in temperature responses across multiple species[32–36], while species distribution modelling at the global scale employs temperature as a key predictor[5]. Similarly, diet breadth and host range are frequently discussed in relation to ecological trade-offs and the evolution of specialization[30,37,38]. Correlations among climatic and microclimatic (i.e., abiotic) niche axes have been analysed in certain vertebrate and plant taxa[39–41]. However, the relationship between abiotic and biotic niche breadth has not hitherto been considered, even in syntheses where data on environmental tolerance and diet breadth have been collated[42]. The popularity of temperature response functions and host ranges in the scientific literature also results in the practical benefit that large quantities of data are available across many species, for analyses such as ours. Continued compilation of ecological traits

for multiple species, such as the TRY database for plants[43], will be key to understanding the relationships between specialization on different niche axes.

We interpreted the smaller $T_{range}$ of in planta processes as compared with in vitro GC as empirical evidence for Hutchinson's distinction between the fundamental niche and the realized niche, where the latter is limited by interactions with other species[3]. Competition and predation are commonly cited as the biotic interactions restricting the size of the fundamental niche[3]. In our case we propose that interactions within the host plant could restrict the performance of the pathogen at high and low temperatures. However, we are unable to speculate on the cellular or biochemical mechanisms which underpin the restricted $T_{range}$ during in planta growth. Empirical tests of the relative size of the realized niche compared with the fundamental niche are rare. One example, however, reconstructed the fundamental temperature niche of amphibians and reptiles from physiological measures of lower and upper critical temperatures, while observed geographical distributions and mean monthly temperature data were used to estimate the realized temperature niche[36]. Here, we have estimated both the fundamental and realized niches from physiological data, because measurements of DD and IN rates incorporate biotic interactions directly. However, we acknowledge limitations in our analysis due to reporting in our data source[15]. Cardinal temperatures for GC were mostly unambiguously tabulated, whereas data for DD and IN were, in comparison, more often described within prose (see Methods for further details).

Invasive fungi and oomycetes are spreading rapidly around the world to challenge global food security, partly in response to climate change[6]. Our analysis suggests that evolution and hence flexibility of temperature responses, and host ranges may both simultaneously contribute to the spread, invasion and threat of destructive plant pathogens. Therefore, the shape and size of the microbial niche has important implications for the management of natural and agricultural ecosystems.

## Methods

**Cardinal temperature data collection**. Minimum ($T_{min}$), optimum ($T_{opt}$) and maximum ($T_{max}$) temperatures (collectively 'cardinal temperatures') of five life-cycle processes (DD, FR, IN, SG and SP), as well as GC (collectively 'biological processes'), were extracted and thence digitized from ref. [15] for fungi and oomycete species. Biological processes such as wood decay, spore discharge, enzyme production, and saltation were excluded due to paucity of data. This dataset is hereafter referred to as the 'Togashi dataset' (see Data availability). In brief, ref. [15] is a compilation of published literature regarding plant pathogen temperature relations, published (in print) in 1949. Ref. [15] contains over 300 pages of data from over 1000 publications (published in the 19th and 20th century). To our knowledge ref. [15] has not previously been digitized. This publication contains data hitherto poorly accessible to the scientific community, and which has not been rigorously interrogated. Additionally, GC $T_{opt}$ and $T_{max}$ data were extracted for 107 *Phytophthora* species from ref. [20], hereafter referred to as the 'Martin dataset'. Finally, IN cardinal temperature for 44 plant pathogen species were extracted from ref. [44], hereafter referred to as the 'Magarey dataset'.

The Index Fungorum (IF) and associated Species Fungorum (SF) databases (www.indexfungorum.org; www.speciesfungorum.org) were used to identify current synonyms for each species recorded in the Togashi dataset (accessed between 8/5/2020 and 15/5/2020). Where no current name was available, or species authorship name(s) were very inconsistent (i.e. no similarity to that cited in ref. [15]), the Mycobank database (www.mycobank.org) was used as an alternative (accessed between 8/5/2020 and 15/5/2020). Species were recorded by their current name, according to the IF/SF or Mycobank databases (as above). Where no current name was explicitly provided, but the species could be identified, the species name searched and located was assumed to be current and correct. However, such records were treated as ambiguous. Similarly, if a species could be identified but species authorship name(s) cited in ref. [15] showed no similarity to that on the IF/SF and/or Mycobank databases, or where species authorship name(s) were not provided by ref. [15], the species was included but treated as ambiguous. Where spelling of species names differed between ref. [15] and the IF/SF and/or Mycobank databases, but it was possible that the spelling in ref. [15] was an error, species names were updated to reflect this, but treated as ambiguous. If a species cited in ref. [15] could not be identified at all on the IF/SF or Mycobank databases (i.e. where no

synonymous names assigned by ref. [15] could be identified), it was excluded from the dataset, except in a few cases, where alternative resources were used to cross-reference species names (see Togashi dataset for further details). In some cases, species were recorded in ref. [15] under multiple synonyms. However, said species were at times found to be not synonymous, once identified in the IF/SF and/or Mycobank databases. Various methods were used to correct for this, detailed below.

If cardinal temperature data in ref. [15] specifically referred to one of the nonsynonymous species, these were recorded under the currently designated name for that species on the IF/SF or Mycobank databases (as explained above). For example, GC cardinal temperature data were recorded for *Fusarium sambucinum* (Fuck.) [syn. *Fusarium discolour* var. *sulphureum* (App. et Wr.), *Fusarium polymorphum* (Mart.), *Fusarium roseum* (Lk.), *Fusarium sulphureum* (Schlecht)]. The IF/SF database classified *F. sambucinum* as *F. roseum* Link (1890) but *F. sulphureum* as *F. sulphureum* Schltdl. (1824). However, a subset of cardinal temperature data in ref. [15] were recorded "as *sulphureum*", and so were assigned to *F. sulphureum*, and not *F. roseum* in the Togashi dataset. In contrast, if cardinal temperature data associated with multiple, nonsynonymous species did not explicitly specify which species the data referred to, two alternative methods were used for clarification. First, the titles of publications cited in ref. [15] for that species record were cross-referenced, to determine if a species name (or disease name that likely suggested a species) was provided in the title. If so, species names were corrected to match that of the publication title, for that data point. For example, GC cardinal temperature data were recorded by ref. [15] for *Corticium vagum* (Berk. et Curt.) [syn. *Rhizoctonia solani* (Kühn)]. The IF/SF databases classified the former as *Botryobasidium vagum* ((Berk. & M.A. Curtis) D.P. Rogers (1935)), but the latter as *R. solani* (J.G. Kühn, (1858)). A subset of titles from publications used by ref. [15] to extract cardinal temperature data contained the term "*Rhizoctonia solani*", and so such data were here assigned to *R. phaseoli*, and not *B. vagum*. However, if no usable information was provided in publication titles, data were here recorded under the first species given by ref. [15]. This name was chosen as it was the bold, title name given to that species record in ref. [15]. For example, in ref. [15] GC cardinal temperature data were recorded for *Fusarium redolens* (Wr.) [syn. *Fusarium reticulatum* (Mont.); *Fusarium spinaciae* (Sherb.)]. The IF/SF databases classified neither *F. reticulatum* nor *F. spinaciae* as synonymous with *F. redolens*. However, it was not clear in ref. [15] which cardinal temperatures referred to which species, and titles of publications used to extract cardinal temperature data by ref. [15] only stated "Fusarium". Hence, all data were here recorded under the bold, title species name—*F. redolens*. Any records that underwent additional processing outlined here were also deemed ambiguous. Any synonyms assigned by ref. [15] that could not be identified were deemed nonsynonymous. Where species in ref. [15] were recorded under multiple species names, that were here found to be synonymous, we assumed all publications used by ref. [15] to extract cardinal temperature data refer only to these species.

All cardinal temperature data concerning *Fusarium oxysporum formae speciales* were recorded under *F. oxysporum*, as well as their respective *formae speciales*. These *formae speciales* were excluded from analyses of within-species cardinal temperature analyses (Fig. 2, Supplementary Tables 2–4, 6, 7), but were included in analyses of niche co-specialization (Fig. 3, Supplementary Tables 5, 8–10), thereby maintaining specific known pathogen–host interactions in the latter analysis.

The methods of determining fungi and oomycete species names resulted in cardinal temperature data for 695 microbes (631 fungi and 64 oomycetes, $N = 8656$) being recorded in the Togashi dataset. Previous analyses of thermal responses have considered only a handful of fungi and no oomycetes[35,45,46]. When data of ambiguous species records (explained above) were excluded, 568 microbes (514 fungi and 54 oomycetes, $N = 6045$) remained in the Togashi dataset. Excluding ambiguous species records had little influence on our results (Supplementary Tables 6, 8). All information regarding how species were named in the Togashi dataset, including species authorship name(s) cited in ref. [15] and the various databases detailed above, changes to spelling of species names cited in ref. [15], apparent synonymous and nonsynonymous species names cited in ref. [15], cases where data were extracted from one species record and recorded as a different species, and species records treated as ambiguous, can be found in the Togashi dataset.

For each data point recorded in the Togashi dataset, where ref. [15] recorded that the true value lies above or below the value provided, the value provided was recorded. For example, if $T_{min}$ was recorded as 'below 8 °C', 8 °C was recorded as $T_{min}$; if $T_{max}$ was recorded as 'above 25 °C', 25 °C was recorded as $T_{max}$. Where a range was provided, the mid-point was recorded. However, where a range was provided, but the true value was recorded to lie above or below this, the upper or lower limit was chosen, respectively. For example, if $T_{min}$ was quoted as 'below 18–20 °C', 18 °C was recorded as $T_{min}$. Where a range was quoted for the entire biological process, the upper and lower bounds were recorded as $T_{max}$ and $T_{min}$, respectively. For example, if IN was quoted as 'occurring between 5 and 35 °C', 5 °C was recorded as $T_{min}$ and 35 °C was recorded as $T_{max}$. However, in cases where it was likely that the temperature range quoted referred to a range of optimal conditions, the mid-point was recorded as $T_{opt}$ unless stated otherwise. Cardinal temperatures were also estimated from prose in ref. [15]. Data under 'IN and DD' were independently recorded under IN and DD, unless the text specifically indicated one of these processes. Data recorded as 'Specialization and resistance' were recorded under IN and/or DD, where appropriate. Data quoted in ref. [15] that

were the result of complex treatments and/or were not likely related to $T_{min}$, $T_{opt}$, or $T_{max}$ were excluded. Further information regarding how each cardinal temperature data point in the Togashi dataset was determined from information provided in ref. [15] is reported in the Togashi dataset. Where multiple references were provided for a single data point in ref. [15], this was taken to represent independent observations, and so were individually included in the Togashi dataset. All data extraction was completed by the same researcher. For the Magarey dataset, cardinal temperatures were recorded as point estimates and pathogen names were updated according to the IF/SF database or Mycobank database (accessed between 8/5/2020 and 15/5/2020) to ensure correct matching to the Togashi dataset for data validation (see below). Finally, for each data point recorded in the Martin dataset, where ref. [20] recorded that the true value lies above or below the value provided, the value provided was recorded, and where a range was provided, the mid-point was recorded. To ensure maximum matching to *Phytophthora* species phylogenies (detailed below), *Phytophthora katsurae* was renamed *Phytophthora castaneae* in the Martin dataset. We also assumed that *Phytophthora ipomoea* corresponded to *Phytophthora ipomoeae*.

**Data analysis.** All analyses were performed in R 3.5.3[47]. In all analyses the means of $T_{min}$, $T_{opt}$ or $T_{max}$ for a given biological process, for a given species, were treated as a single data-point. Where more than five related statistical tests were conducted (Supplementary Tables 2, 5, 7, and 10) the Holm–Bonferroni correction[48] for multiple tests was applied, with adjusted significance levels given in table legends.

**Data validation.** Sixteen pathogens were recorded in both the Togashi and Magarey datasets. For these species, root mean square error (RMSE) was calculated between IN cardinal temperature estimates. When all data were included, RMSE was 5.15 °C ($N = 43$) (Supplementary Fig. 3a). Clustering of data points at 35 and 1 °C along the y-axis is a result of how ref. [44] estimated $T_{max}$ and $T_{min}$, respectively —if no $T_{max}$ for IN was found, the authors set $T_{max}$ to 35 °C. Similarly, if no $T_{min}$ for IN was found, but IN could occur lower than the hosts developmental threshold, the authors set $T_{min}$ to be 5 °C lower than the lowest tested temperature, but not lower than 1 °C. When $T_{min}$ data recorded as 1 °C and $T_{max}$ data recorded as 35 °C in the Magarey dataset were excluded, RMSE was 4.73 °C ($N = 29$) (Supplementary Fig. 3b). The greatest deviation from an identity relationship (dotted line) occurred around $T_{min}$ (Supplementary Fig. 3a, b). This may be due to $T_{min}$ being more problematic to quantify—the lowest temperature a given biological process occurs at will depend on the amount of time given for the process to occur. $T_{max}$ is likely to be more clearly defined as cells will die at high temperature. Further, 22 *Phytophthora* species were present in both the Togashi and Martin datasets. For these pathogens, RMSE was calculated for GC $T_{opt}$ as 2.65 °C ($N = 20$) (Supplementary Fig. 3c) and GC $T_{max}$ as 3.34 °C ($N = 22$) (Supplementary Fig. 3d). Where multiple, independent cardinal temperature estimates were cited for the same species the mean was taken for all analyses above. Abstracting cardinal temperatures for GC from ref. [15] was straightforward because data were mostly tabulated. In contrast, DD and IN data in ref. [15] were more often written in prose (see the Togashi dataset for further details). This is one possible explanation for the greater calculated RMSE for IN than GC.

**Analysis of cardinal temperature.** The Togashi dataset was used for this analysis. Where multiple, independent cardinal temperature estimates were cited for the same species and biological process in ref. [15], the mean was taken. Supplementary Data 1 provides summary information regarding species–biological process–cardinal temperature sample sizes for the Togahsi dataset. $T_{range}$ was calculated as the range between $T_{min}$ and $T_{max}$. $T_{range0.5}$ was calculated as the range between $T_{min0.5}$ and $T_{max0.5}$; $T_{max0.5}$ and $T_{min0.5}$ refer to $T_{max}$ and $T_{min}$ where a species response rate = 0.5 (at $T_{opt}$ the responses = 1, at $T_{min}$ and $T_{max}$ the response = 0). Hence, $T_{range0.5}$ reflects the temperature range where a species performs a biological process well. Responses were calculated by a beta function (Eq. (1)) that uses a species' cardinal temperature to estimate a temperature performance curve[16]. Skew was calculated according to Eq. (2), Where skew >0.5, $T_{opt}$ is closer to $T_{max}$ than $T_{min}$; where skew <0.5, $T_{opt}$ is closer to $T_{min}$ than $T_{max}$. Species with at least one $T_{opt}$, $T_{range}$ or skew estimate were included in analyses involving $T_{opt}$, $T_{range}$ and skew, respectively.

$$r(T) = \left(\frac{T_{max} - T}{T_{max} - T_{opt}}\right)\left(\frac{T - T_{min}}{T_{opt} - T_{min}}\right)^{(T_{opt} - T_{min})/(T_{max} - T_{opt})} \quad (1)$$

$$\text{skew} = \frac{T_{opt} - T_{min}}{T_{max} - T_{min}} \quad (2)$$

In some cases, for particular species–biological process combinations, mean $T_{opt}$ was estimated as greater than mean $T_{max}$ or lower than mean $T_{min}$. This is because data from multiple, independent sources were provided within ref. [15]. For such cases, nonsensical values (i.e. skew <0 or >1) were removed for these species–biological process combinations. Differences between cardinal temperatures for GC and other processes were compared within species using two-sided $t$-tests (Supplementary Table 2). Association between GC $T_{opt}$ and $T_{opt}$ of other biological processes was investigated using two-sided Pearson correlation

(Supplementary Table 3). The same analysis was performed for $T_{range}$ (Supplementary Table 4). Sample size varies in the Togashi dataset as a species may have a $T_{min}$, $T_{opt}$ and/or $T_{max}$ estimate for one biological process, but not others.

**Niche co-specialization.** The Togashi dataset was used for this analysis. The Plantwise database (CABI) (accessed 28/10/2013, by permission) provides information on known pathogen/host interactions. To improve matching of pathogen species between the Togashi dataset and the Plantwise database, 85 pathogen species names were updated in the Plantwise database (Supplementary Table 11), according to their respective, current names given in the IF/SF and/or Mycobank databases [accessed between 8/5/2020 and 15/5/2020]. As above, Mycobank was used where no information was available on IF/SF. Species authorship names are not recorded in the Plantwise database and so were not considered here. Hence, it was assumed that if any alternative current name for a pathogen in the Plantwise database was present in the Togashi dataset, it was a correct match. *Sensu* species names recorded in the IF/SF and Mycobank databases were also included during this matching process. Authorship names of current species were cross-checked to the Togashi database, to ensure current species were a true match.

All recorded plant hosts of fungi and oomycetes included in the Togashi dataset were identified. Host variety was not considered (i.e. hosts were recorded no further than species rank). *Peronospora farinosa* was assigned all hosts recorded for *P. farinosa*, as well as those recorded for *P. farinose formae speciales* in the Plantwise database. Similarly, *F. oxysporum* was assigned all hosts recorded for *F. oxysporum*, as well as those recorded for *F. oxysporum formae speciales*. *F. oxysporum formae speciales* were also included in this analysis as individual data points due to *formae speciales* cardinal temperature data available in the Togashi dataset. Two different methods were used to quantify host diversity of pathogens. First, only hosts recorded to species level in the Plantwise database were included. In this case, 1016 hosts of 302 pathogens were utilized to generate a time-calibrated host phylogeny using the R function 'S.PhyloMaker' (scenario 1, genera or species added as basal polytomies within their families or genera)[49]. The resultant generated host phylogeny is hereafter referred to as the 'unprocessed host phylogeny' (Supplementary Fig. 4). Second, where a host record in the Plantwise database was not identified to species, it was assumed that the pathogen in question was able to successfully infect all species present in S.PhyloMaker, within the taxonomic rank reported. For example, *Macrophomina phaseolina* was recorded in the Plantwise database as being a pathogen of the class Pinopsida. Hence, 419 host species found within the class Pinopsida in S.PhyloMaker were added to *M. phaseolina* host range. S.PhyloMaker did not report above family classification. Hence, we assumed that pathogens reported to infect class Pinopsida included the families Araucariaceae, Cephalotaxaceae, Cupressaceae, Pinaceae, Podocarpaceae, Sciadopityaceae, and Taxaceae, and to infect order Gentianales included the families Apocynaceae, Gelsemiaceae, Gentianaceae, Loganiaceae, and Rubiaceae. In this case, 15,982 hosts of 309 pathogens were used to generate a time-calibrated host phylogeny, also using the R function 'S.PhyloMaker' (scenario 1)[49]. The resultant generated host phylogeny is hereafter referred to as the 'processed host phylogeny' (Supplementary Fig. 5). To improve correct matching of plant host species names to S.PhyloMaker or improve positioning of species during phylogeny construction, some corrections to host species names in the Plantwise database were made, according to The Plant List (TPL) (www.theplantlist.org) (accessed between 19/3/2020 and 16/5/2020) (Supplementary Table 12). First, hosts species in the Plantwise database that were not identifiable to genus-level in S.PhyloMaker were corrected, where possible. Second, hosts not identifiable to species-level in S. PhyloMaker were corrected, where possible. This method ensured that during phylogeny construction (1) all hosts species included in the analysis were identifiable to at least genus-level in S.PhyloMaker and (2) we maximized the number of host species identified to species-level. In all cases, author or publication details of host species names was not considered, as this information was not provided in the Plantwise database. Hence, if multiple accepted names were provided by TPL, the following method was applied. First, if any of the accepted species name given by TPL were identical to that given by the Plantwise database, this name was given. Second, if none of the accepted species names given by TPL were identical to the Plantwise database, a single TPL-accepted species name was selected at random for that host. Seven hosts species (*Abelmoschus esculentus*, *Cyphomandra betacea*, *Cuprocyparis leylandii*, *Elettaria cardamomum*, *Gloriosa rothschildiana*, *Coleus forskohlii*, and *Ullucus tuberosus*) as well as the genus-level records *Ascocenda*, *Elettaria*, and *Scindapsus* were not identifiable to genus-level in S.PhyloMaker, and hence were excluded from the analysis. This was due to uncertainty in classification and phylogenetic position, or seemingly missing data in S.PhyloMaker. Further information is provided in Supplementary Table 9.

The function 'pd' in the R package 'picante'[50] was used to quantify host diversity of each pathogen. Host diversity was calculated as Faith's phylogenetic diversity (PD)[51]. The phylogeny root node was excluded in all calculations. Hence, pathogens with a single host were assigned a PD of zero. Fewer pathogens were included for analyses involved $T_{range0.5}$ as this parameter required estimates of $T_{min}$ and $T_{max}$, as well as $T_{opt}$. Co-specialistion across abiotic ($T_{range}$ or $T_{range0.5}$) and biotic ($\log_{10+1}$-transformed host diversity) niche axes was calculated by two-sided Pearson correlation.

**Cardinal temperature phylogenetic signal**. The Martin dataset was used for this analysis. Phylogenies constructed by (1) Bayesian, (2) maximum likelihood, and (3) maximum parsimony methods for *Phytophthora* species were extracted from ref. [19] (TreeBASE S19303). 101 *Phytophthora* species (*P. alni* ($T_{opt}$ and $T_{max}$ calculated as the average of *P. alni* sub. sp. *alni*, *P. alni* sub. sp. *multiformis*, and *P. alni* sub. sp. *uniformis* in ref. [20]), *P. alticola, P. andina, P. aquimorbida, P. arenaria, P. austrocedrae, P. bisheria, P. boehmeriae, P. botryose, P. brassicae, P. cactorum, P. cajani, P. cambivora, P. capensis, P. capsici, P. captiosa, P. castaneae, P. chrysanthemi, P. cinnamomi, P. citricola, P. citrophthora, P. clandestina, P. colocasiae, P. constricta, P. cryptogea, P. drechsleri, P. elongata, P. erythroseptica, P. europaea, P. fallax, P. fluvialis, P. foliorum, P. fragariae, P. frigida, P. gallica, P. gemini, P. gibbosa, P. glovera, P. gonapodyides, P. gregata, P. hedraiandra, P. heveae, P. hibernalis, P. humicola, P. hydropathica, P. idaei, P. ilicis, P. infestans, P. inflata, P. insolita, P. inundata, P. ipomoeae, P. iranica, P. irrigata, P. kernoviae, P. lateralis, P. litoralis, P. macrochlamydospora, P. meadii, P. medicaginis, P. megakarya, P. megasperma, P. melonis, P. mengei, P. mexicana, P. mirabilis, P. morindae, P. multivesiculata, P. multivora, P. nemorosa, P. nicotianae, P. obscura, P. palmivora, P. parsiana, P. phaseoli, P. pini, P. pinifolia, P. pistaciae, P. plurivora, P. polonica, P. primulae, P. pseudosyringae, P. pseudotsugae, P. psychrophila, P. quercetorum, P. quercina, P. quininea, P. ramorum, P. richardiae, P. rosacearum, P. rubi, P. sansomeana, P. siskiyouensis, P. sojae, P. syringae, P. tentaculata, P. thermophila, P. trifolii, P. tropicalis, P. uliginosa,* and *P. vignae*) were present in both the Martin dataset and extracted phylogenies. We assumed that *Phytophthora* x *alni* recorded in ref. [19] corresponded to *Phytophthora alni*. The function 'phylosig' in the R package 'phytools'[52] was used to separately test for a phylogenetic signal for GC $T_{opt}$ and $T_{max}$ ($N = 101$). 10,000 simulations were run in each analysis for randomization test. Where multiple strains of a particular *Phytophthora* species were included in a phylogeny, only one strain was assigned a GC $T_{opt}$ or $T_{max}$ record from the Martin dataset, thereby preventing pseudoreplication.

The influence of spatial autocorrelation on phylogenetic signal of *Phytophthora* species cardinal temperature was investigated (Supplementary Fig. 1). For 31 *Phytophthora* species included in the above analysis (*P. alni, P. boehmeriae, P. botryosa, P. cactorum, P. cambivora, P. capsici, P. castaneae, P. cinnamomi, P. citrophthora, P. colocasiae, P. cryptogea, P. drechsleri, P. erythroseptica, P. fragariae, P. infestans, P. kernoviae, P. lateralis, P. macrochlamydospora, P. meadii, P. medicaginis, P. megakarya, P. megasperma, P. nicotianae, P. palmivora, P. pseudosyringae, P. quercetorum, P. quercina, P. ramorum, P. rubi, P. sojae,* and *P. vignae*), estimates of presence at country or region scale were extracted from CABI Plantwise[53,54]. To maximize species matching between datasets, *P. erythroseptica* var. *erythroseptica* was renamed *P. erythroseptica*, *P. drechsleri* f.sp. *cajani* was renamed *P. drechsleri*, and *P. katsurae* was renamed *P. castaneae* in the Plantwise database.

The centroid of the country or region were used for all records. Mantel correlations (MCs) were performed between GC $T_{opt}$ (and $T_{max}$) distance and great circle distance (km) or average air surface temperature (AST) distance (ºC) matrices. Gridded average AST (January 1951 and December 1980) was extracted from Berkley Earth (www.berkeleyearth.org) (accessed 19/11/2017) for each latitude–longitude location. Latitude and longitudes values in the CABI Plantwise database were rounded to the nearest 1° interval, to align with those extracted from Berkley Earth for analysis of average air surface temperature distance.

All MC were performed using the function 'mantel' in the R package 'ecodist'[55] with 10,000 iterations to calculate bootstrapped confidence limits. *P* values were calculated according to a null hypothesis that MCs were equal to zero (two-tailed test).

**Co-phylogenetic association**. The Martin dataset was used for this analysis. 35 *Phytophthora* species were present in both the Plantwise database and extracted *Phytophthora* phylogenies detailed above (*P. alni, P. asparagi, P. boehmeriae, P. botryose, P. cactorum, P. cambivora, P. capsica, P. castaneae, P. cinnamomi, P. citricola, P. citrophthora, P. colocasiae, P. cryptogea, P. drechsleri, P. erythroseptica, P. fragariae, P. hibernalis, P. infestans, P. kernoviae, P. lateralis, P. macrochlamydospora, P. meadii, P. medicaginis, P. megakarya, P. megasperma, P. nicotianae, P. palmivora, P. phaseoli, P. pseudotsugae, P. ramorum, P. richardiae, P. rubi, P. sojae,* and *P. syringae*). To maximize species matching between datasets, *P. erythroseptica* var. *erythroseptica* was renamed *P. erythroseptica*, *P. drechsleri* f.sp. *cajani* was renamed *P. drechsleri*, and *P. katsurae* was renamed *P. castaneae* in the Plantwise database. As previously, we also assumed that *Phytophthora* x *alni* recorded in ref. [19] corresponded to *Phytophthora alni*. The 258 hosts of these pathogens recorded to species level in the Plantwise database were extracted (i.e. those only recorded to genus or family were excluded, and host variety was not considered) and utilized to generate a time-calibrated host phylogeny using the R function 'S.PhyloMaker' (scenario 1) (Supplementary Fig. 6). As above, to improve correct matching of plant species names to S.PhyloMaker, some corrections to host species names in the Plantwise database were made, according to The Plant List (TPL) (www.theplantlist.org) (accessed 19/3/2020) (Supplementary Table 12). 239 hosts matched to species level in S.PhyloMaker and 17 hosts matched to genus level. Two hosts (*C. betacea* and *E. cardamomum*) were not identifiable to genus-level in S.PhyloMaker, and hence were excluded from the analysis. The function 'PACo' in the R package 'paco'[56] was used to test for co-phylogenetic association between each *Phytophthora* species phylogeny and the generated host phylogeny ($N = 35$). We applied a square root correction to the patristic distance matrices calculated from the *Phytophthora* phylogenies due to negative eigenvalues[57]. Host and pathogen phylogenies were standardized prior to super-imposition, resulting in the best-fit of the superimposition being independent of both phylogenies[57]. Additionally, the method quasiswap was assigned, which is a more constrained method than others available, where the number of interactions is conserved for each species (and hence in the network as a whole). These methods were chosen because we make no assumption about which group (host or pathogen) is tracking the other[57]. 10,000 randomizations were run in each analysis. Under perfect co-phylogenetic association, the best-fit Procrustean super-imposition ($m^2_{XY}$) is zero. As co-phylogenetic association declines, $m^2_{XY}$ tends towards that calculated in the ensemble of network randomizations in each null model. Where multiple strains of a particular *Phytophthora* species were included in a phylogeny, only one strain was assigned a host range, thereby preventing pseudo-replication.

**Influence of uncertainty in reported cardinal temperatures**. Cardinal temperature data in the Togashi and Martin datasets contain uncertainties due to reported values varying from their true values (i.e. $T_{max} < 32$). We investigated whether these potential uncertainties could have affected our conclusions concerning fundamental vs. realized niche geometry, niche cospecialisation, and cardinal temperature phylogenetic signal. It was beyond the scope of this study to establish how cardinal temperatures were determined for each record reported in ref. [15]. Further, ref. [20] does not provide information or references as to how reported $T_{opt}$ and $T_{max}$ were determined. To overcome this, we assumed that in all cases cardinal temperature was investigated experimentally at 5 °C increments, and that the minimum and maximum temperature treatments spanned $T_{range}$. This implies that on average cardinal temperature estimates do not deviate from their true values by more than ±2.5 °C. Hence, for both the Togashi and Martin datasets, 2.5 °C was added or subtracted from cardinal temperature data reported as being above or below their true value, respectively. For example, GC data reported as >25 °C was modified to 27.5 °C. Where data in the Togashi dataset were extracted from prose, data were modified in this way only if we could determine the direction of the error with confidence. In all cases, errors have little effect on our results and did not affect any of our key conclusions (Supplementary Tables 7, 10, and 13). This suggests that uncertainties are randomly distributed and do not affect the results presented here.

**Reporting summary**. Further information on research design is available in the Nature Research Reporting Summary linked to this article.

## Data availability

The Togashi dataset is available on the Dryad repository with identifier doi:10.5061/dryad.tqjq2bvw6. Additional temperature response data are available from refs. [20,44]. Fungal and oomycete host plant data and geographical distributions (the Plantwise database) were used under license for the current study, and are available with permission from CABI, Wallingford, UK. Fungi and oomycete taxonomic data are available from the Index Fungorum (www.indexfungorum.org), Species Fungorum (www.speciesfungorum.org) and MycoBank (www.mycobank.org) databases. Plant species names are available from The Plant List (www.theplantlist.org). *Phytophthora* species phylogenies are available from ref. [19]. Gridded average surface air temperature data are available from Berkley Earth (www.berkeleyearth.org). The source data underlying Figs. 1, 2, and 3, and Supplementary Figs. 1 and 2 are provided as a Source Data file. Source data are provided with this paper.

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

## Acknowledgements

Authors thank Angus Buckling for comments on the manuscript and Paul Kirk for providing information on fungi and oomycete genera names. TMC is funded by a BBSRC SWBio DTP studentship (BB/M009122/1). S.J.G. acknowledges a CIFAR Fellowship in Fungal Kingdom: Threats & Opportunities. D.P.B. and S.J.G. are funded by BBSRC grant BB/N020847/1.

## Author contributions

T.M.C. collated and curated the data, analysed the data, created figures and wrote the manuscript. S.J.G. reviewed and edited the manuscript. D.P.B. developed the concept and methodology, supervised the research, analysed data, created figures and wrote the manuscript.

## Competing interests

The authors declare no competing interests.
