## [Peer Review File · Nature Communications]

Reviewers' Comments:

Reviewer #1:

Remarks to the Author:

Review of Chaloner et al. Geometry and evolution of the ecological niche in plant-associated microbes.

This is a really interesting paper that provides the first (to my knowledge) such extensive and complete analysis of compelling set of questions about the niche geometry of plant pathogens. The questions are important because climate change and the redistribution of host species are causing major reshuffling of the distribution of pathogens and their potential effects. Understanding the potential for evolution in situ and migration, and the plasticity of biotic and abiotic niche spaces for pathogens, is a topic that has been discussed, but not tackled in this concrete a way. In fairness, there was a twinge of regret as I realized that a project my collaborators and I began a number of months ago had just been scooped. This was the paper we hoped to write.

I very much like the introduction and framing of the paper. I think this was a very solid approach to explaining the abiotic and biotic niche determinants and why this is an interesting question to address. I found the first several paragraphs very compelling and pushing me to want to know more.

p2 para 2. "This suggests a fundamental difference in the shape of the temperature response for growth in axenic culture (GC) than for processes that involve interaction with the host plant or occur without nutrient media." There are also fundamental differences in the biology of hyphal growth and spore germination that go beyond the axenic culture vs. host plant or media-independent. I am not comfortable with the generalization here. The differences are interesting but whether that has to do with how it is growing/interacting with environment or because of differences between growth and germination cannot easily be deciphered. Please take care of the language used.

Fig 1. I think the figures -- while well structured and interesting -- would be easier to read if the line thickness on the box and whiskers were thinner. They merge together too much and it is hard to read clearly.

p3 para 2. "The terms specialist and generalist therefore cannot be applied as holistic descriptions of fungal or oomycete species' ecology". Yes, this makes sense from the data. But it is not clear to me why one would expect that to be the case. I do not recall any description of a pathogen as being a global generalist that includes both host range and abiotic conditions, or an extreme host specialist that is also an abiotic specialist. In fact, I would expect that being an extreme specialist for both host and abiotic conditions would seldom be an evolutionarily durable strategy. So make it clearer why this is an interesting result.

I very much like and appreciate the phylogenetic analysis of T range in *Phytophthora*. This is a significant and very interesting finding. Even while modest in effect, this is consequential for thinking about how pathogens are likely to respond and how useful patterns derived from well studied species can be applied to less studies species.

Similarly, the discussion of evolutionary lability of host range and abiotic ranges is a valuable contribution. These are questions that have been asked -- and it is great to have the start of some answers. I recognized that this is still first pass on the question, but it is an important step.

p5 para 1. "abiotic Fundamental Niches are both wider and different in form than corresponding

Realized Niches." When are they not?

p5 para 1. "We show that microbial specialization can occur independently in biotic and abiotic niche axes, suggesting that the terms "specialist" and "generalist" should be used cautiously when describing the ecology of microbial species." Again, has anyone actually conflated these? It is useful to note it is the case, but not surprising. Is there more that can be said about the ways in which they are different?

I thought the description of methods was adequate. The details in matching up different datasets, and especially the host range data and nomenclature changes are often daunting, but I think the number of records, the general reliability of the data sources used, and the apparent care was adequate for me to feel comfortable. One question I had was whether there was any assessment of the impact of how data were handled when the cardinal temperatures were "above #C" or "below #C". How much does having estimates that are consistently less extreme than the biological values affect the results?

In the supplementary Tables and Figures -- it might be overkill, but I sometimes found I had to go back to the text to remember what a particular abbreviation meant. It would be helpful to have the captions complete enough so the tables stand alone.

Reviewer #2:

Remarks to the Author:

The manuscript entitled "Geometry and evolution of the ecological niche in plant-associated microbes" provides an interesting opportunity to explore the ecological niche of plant-associated fungal pathogens. Their primary finding is that thermal niches of plant-associated fungal pathogens can occur independently of abiotic or biotic niche axes. This has implications for management of fungal pests in natural and agricultural ecosystems. Overall, I found this manuscript to be thought-provoking, well-written, and a clever use of existing datasets. Below are a few comments to improve clarity and avoid misleading arguments.

1. Is there a significant difference between the skew for temperature response for the fungi/oomycetes grown in the axenic media compared to the other rates measured? It seems like a large portion of your interpretation is based on this skew being different, but I don't see it actually stated anywhere. Can you add P values to Figures 1 and 2?
2. The understanding of temperature response needs to be improved/updated. For example, the paper cited for "Rates increase with temperature to T_{opt} following thermodynamic expectations, followed by rapid decline in rate as enzymes denature" (Schulte 2015), actually includes text challenging the viewpoint that enzymes denature following T_{opt} (see Hobbs et al. 2013, ACS chem biology).
3. The discussion of the phylogenetic components could be improved. Can you calculate the phylogenetic signal for all of the species in the dataset together? It might be that temperature response is highly constrained based on physiological characteristics of the organism/enzymes.
4. It would be helpful to include a discussion of the biases of the dataset. For example, more T_{min} than T_{max} values could correspond to a biased T_{range} value, which is the more important indicator of the thermal niche breadth. The T_{min} and T_{max} values could additionally be biased because temperatures measured are chosen by the experimenters. Also, overabundance of particular type of organism in the analysis could lead to misrepresentation of the results.

5. I don't understand how you can make claims about not responding to biotic or abiotic niche axes when you only test a few components associated with biotic and abiotic niches. Can you de-generalize these types of statements in the manuscript or add additional clarification?

Reviewer #3:

Remarks to the Author:

Chaloner et al has prepared manuscript entitled "Geometry and evolution of the ecological niche in plant-associated microbes," which addresses the timely and exciting topic of niche geometry and specialization across niche axes in plant pathogens. This is a question I have long wondered about and am excited to see investigated. I commend the authors for the strong conceptual and applied motivations they put forth for study. Thanks to the authors for a well-motivated and conceptually engaging framework!

I did have some comments and questions about way in which the authors carried out the work:

I know there is significant limitations on the length of the paper. However, there main text does not cite the data sources. If the methods are not in the printed paper (which maybe they are based on my recollection of other Nature Communications papers), the authors should at a minimum refer the readers to the supplementary methods that explains where that data comes from.

Does the data collected on the thermal niche of the fungi and oomycetes account for within species variation in thermal tolerances? To understand the quality of these printed estimates of cardinal temperatures would require more description of how the data was originally collected. How many data points went into the T_{opt} , T_{max} , and T_{min} of a species? Are these points from identical lab strains or do these estimates account for within species variation among geographically distant strains? Summarizing some of this information about the data early on would help readers understand what is being done.

The authors mention that the two databases were used to identify synonymous names for each species and were accessed between "1/5/2017 - 18/10/2019" This window seems long as by the time it is published it will have been three years. Similarly in the section on host associations later in the methods the authors mention that the data on known pathogen/host interactions from the PlantWise database (CABI) was accessed more than 6 years ago (28/10/2013). This means some of the classifications and host breadths have likely changed, but in my opinion this shouldn't stand in the way of publication as realistically data processing takes time.

I am very concerned about this data collation decision described by the authors as: "If cardinal temperature data associated with multiple, nonsynonymous species did not specify which species the data referred to, data were recorded under the first species given in ref. 30. For example, in ref. 30 GC and DD cardinal temperature data were recorded for *Mycosphaerella tulasnei* (Jancz.) [syn. *Cladosporium herbarum* (Link)]. The SFD classified *Mycosphaerella tulasnei* (Jancz.) Lindau (1903) as *Mycosphaerella tassiana* (De Not.) Johanson (1884), but *Cladosporium herbarum* (Pers.) Link (1816) as *Cladosporium herbarum* (Pers.) Link (1816). However, it was not clear in ref. 30 which cardinal temperatures referred specifically to which species. Hence, all data were recorded under *Mycosphaerella tassiana*." Instead of assigning all the data to the first species that happened to be listed in ambiguous cases in ref 30, these data should have just been excluded since the authors could not determine which species the information was associated with. How many species did this affect? What happens to the results when the ambiguous cases are excluded?

In the methods statement "For the Magarey dataset, pathogen names were updated according to the SFD or Mycobank [accessed between 1/5/2017 - 18/10/2019] to ensure correct matching to the Togashi dataset. No additional processing was performed on the Martin dataset." After the complex set of decisions explained for the Togashi dataset, it leaves the reader wondering why no explanation of decisions of t_{max} , t_{min} , t_{opt} was needed for either of these datasets and why looking up names in SFD/Mycobank, etc wasn't necessary for the Martin dataset. A little more explanation would be ideal.

Authors say "All analyses were performed in R 3.2.336. In all analyses the mean of T_{min} , T_{opt} or T_{max} for a given biological process, for a given species, is treated as a single datapoint." Treating each species as a single data point makes sense, but how did they deal with species with multiple measurements of t_{min} , t_{max} , or t_{opt} ?

Authors wrote in the Analysis of cardinal temperature section and the Niche co-specialisation sections that "The Togashi dataset was used for this analysis." Why only that dataset?

Also later in the paragraph on that analysis the authors say "In some cases, for particular species-biological process combinations, mean T_{opt} was estimated as greater than mean T_{max} or lower than mean T_{min} . This is because data were extracted from multiple sources." I'm confused about what is meant by multiple sources since only Togashi is used for this analysis. Similarly, later still in the section the authors say "Species with at least one T_{opt} , Trange or skew estimate were included in analyses involving T_{opt} , Trange and skew, respectively." What is meant by "at least one"? Does this mean that in some cases species have multiple measurements of T_{opt} or of Trange? I suspect a lot of this confusion would be circumvented with more explanation of the dataset.

What does the '(224)' in the following sentence mean? "235 (224) pathogens of hosts included in the processed host phylogeny were identified with at least one Trange (Trange(50%)) estimate, for at least one biological process, in the Togashi dataset (Supplementary Table 7)."

In the supplementary figure1 on the "Analysis of spatial correlation on phylogenetic signals calculated for Phytophthora species cardinal temperatures", the caption mentions that the p-values for the mantel correlations are all less than <0.05 . Then the authors say that "The Mantel correlations are near zero hence can ignore the influence of spatial effects on our analysis of cardinal temperature phylogenetic signal". I may be misunderstanding the meaning of this statistical test, but this statement that we can ignore spatial effects seems incongruent with the significant p-values.

Supplementary Figure 3 is especially helpful.

**Geometry and evolution of the ecological niche in plant-associated microbes**

Chaloner TM, Gurr SG & Bebbler DP

**Response to reviewers' comments**

We thank the three reviewers for their useful and insightful inputs and suggestions. Their comments
have led us to add detail to the methods, particularly on microbial nomenclature, and to reappraise the
complete Togashi data set. We believe this has further improved the manuscript. Amendments in the
resubmitted manuscript are highlighted in yellow.

**Reviewer 1**

This is a really interesting paper that provides the first (to my knowledge) such extensive and
complete analysis of compelling set of questions about the niche geometry of plant pathogens. The
questions are important because climate change and the redistribution of host species are causing
major reshuffling of the distribution of pathogens and their potential effects. Understanding the
potential for evolution in situ and migration, and the plasticity of biotic and abiotic niche spaces for
pathogens, is a topic that has been discussed, but not tackled in this concrete a way. In fairness, there
was a twinge of regret as I realized that a project my collaborators and I began a number of months
ago had just been scooped. This was the paper we hoped to write. I very much like the introduction
and framing of the paper. I think this was a very solid approach to explaining the abiotic and biotic
niche determinants and why this is an interesting question to address. I found the first several
paragraphs very compelling and pushing me to want to know more.

We thank the reviewer for their very kind and supportive comments, we are delighted that the
manuscript was of interest.

p2 para 2. "This suggests a fundamental difference in the shape of the temperature response for
growth in axenic culture (GC) than for processes that involve interaction with the host plant or occur
without nutrient media." There are also fundamental differences in the biology of hyphal growth and
spore germination that go beyond the axenic culture vs. host plant or media-independent. I am not
comfortable with the generalization here. The differences are interesting but whether that has to do
with how it is growing/interacting with environment or because of differences between growth and
germination cannot easily be deciphered. Please take care of the language used.

Growth in culture (GC) and spore germination (SG) actually have similar T_{range} , it is the host-
interaction processes (disease development, infection, etc.) that have smaller T_{range} . We interpret this
in the light of ecological theory relating to the breadth of the Fundamental vs. Realized Niches, where
the Realized Niche is restricted by biotic interactions. We appreciate that this does not address the
biological mechanism for this restriction. We have included further discussion on our interpretation,
including the lack of knowledge of a mechanism driving the reduced T_{range} for *in planta* vs *in vitro*
growth (lines 207 – 224).

Fig 1. I think the figures -- while well structured and interesting -- would be easier to read if the line
thickness on the box and whiskers were thinner. They merge together too much and it is hard to read
clearly.

We have amended line widths in the plots for increased clarity.

p3 para 2. "The terms specialist and generalist therefore cannot be applied as holistic descriptions of
fungal or oomycete species' ecology". Yes, this makes sense from the data. But it is not clear to me
why one would expect that to be the case. I do not recall any description of a pathogen as being a
global generalist that includes both host range and abiotic conditions, or an extreme host specialist
that is also an abiotic specialist. In fact, I would expect that being an extreme specialist for both host
and abiotic conditions would seldom be an evolutionarily durable strategy. So make it clearer why
this is an interesting result.

The ecological literature has never differentiated specialism on biotic and abiotic niche axes. This is
why we believe this to be an important result. Classic texts (e.g. Chase, J. M., & Leibold, M. A. 2003.
*Ecological Niches: Linking Classical and Contemporary Approaches*. University of Chicago Press),
ecological textbooks (e.g. Begon, Harper & Townsend), studies (Forister, M. L., et al. 2015. The
global distribution of diet breadth in insect herbivores. *Proceedings of the National Academy of*
*Sciences*, 112, 442–447) and reviews (e.g. Sexton, J. P. et al. 2017. Evolution of Ecological Niche
Breadth. *Annual Review of Ecology, Evolution, and Systematics*, 48, 183–206) of niche breadth refer
to ‘specialists’ and ‘generalists’, under the implicit assumption that these ecological strategies apply
to the n-dimensional niche as a whole. We have identified a small number of studies that have
investigated climatic niche breadth correlation, and we have cited these in the Discussion. Ours is the
first (to our knowledge) study that explicitly investigates niche breadth on biotic vs. abiotic axes.

Colloquially, generalists are often referred to as “Jack of all trades, master of none”. Our analysis
reveals that many species can be described as “Jack of *some* trades, master of *others*”. We believe this
is a new and hitherto unexplored description of the shape of the n-dimensional ecological niche.

I very much like and appreciate the phylogenetic analysis of T range in *Phytophthora*. This is a
significant and very interesting finding. Even while modest in effect, this is consequential for thinking
about how pathogens are likely to respond and how useful patterns derived from well studied species
can be applied to less studied species.

Similarly, the discussion of evolutionary lability of host range and abiotic ranges is a valuable
contribution. These are questions that have been asked -- and it is great to have the start of some
answers. I recognized that this is still first pass on the question, but it is an important step.

We thank the reviewer for these very supportive comments.

p5 para 1. "abiotic Fundamental Niches are both wider and different in form than corresponding
Realized Niches." When are they not?

Ecological theory predicts that the Realized Niche is a subset (or restriction) of the Fundamental
Niche. This hypothesis has rarely been tested, and not (to our knowledge) in microbes. Hence, our
results are a test, and validation, of this hypothesis.

p5 para 1. "We show that microbial specialization can occur independently in biotic and abiotic niche
axes, suggesting that the terms “specialist” and “generalist” should be used cautiously when
describing the ecology of microbial species." Again, has anyone actually conflated these? It is
useful to note it is the case, but not surprising. Is there more that can be said about the ways in which
they are different?

As no study has previously investigated the degree of co-specialization across biotic vs abiotic niche
axes, we do strongly feel that this is a surprising and novel result (please refer to earlier response).
However, we would be very interested to cite earlier work undertaking an empirical investigation of
this kind, if available.

I thought the description of methods was adequate. The details in matching up different datasets, and
especially the host range data and nomenclature changes are often daunting, but I think the number of
records, the general reliability of the data sources used, and the apparent care was adequate for me to
feel comfortable.

We thank the reviewer for their positive assessment of our analyses. Please also see replies to
comments by Reviewer 3 on how we have further improved our methodology and analyses.

One question I had was whether there was any assessment of the impact of how data were handled
when the cardinal temperatures were "above #C" or "below #C". How much does having estimates
that are consistently less extreme than the biological values affect the results?

We acknowledge that imprecise cardinal temperature estimates add error to our analyses. It is likely
that cardinal temperature values are quoted as “above #C” or “below #C” in Togashi (1949) or Martin
(2012) because no activity was detected at the next temperature tested above or below this,
respectively. The size of the experimental temperature steps is often unknown. We have however

clearly stated in the Togashi database which data points suffer this limitation (as well as
comprehensive information of how other data points were determined). We do not provide this for the
Martin database, this information is readily available from Martin (2012). We have also provided
additional discussion regarding limitations in our analysis due to reporting in our data sources (lines
102 221 – 224, 374 – 377).

In the supplementary Tables and Figures -- it might be overkill, but I sometimes found I had to go
back to the text to remember what a particular abbreviation meant. It would be helpful to have the
captions complete enough so the tables stand alone.

This has been addressed.

**Reviewer #2 (Remarks to the Author):**

The manuscript entitled “Geometry and evolution of the ecological niche in plant-associated
microbes” provides an interesting opportunity to explore the ecological niche of plant-associated
fungal pathogens. Their primary finding is that thermal niches of plant-associated fungal pathogens
can occur independently of abiotic or biotic niche axes. This has implications for management of
fungal pests in natural and agricultural ecosystems. Overall, I found this manuscript to be thought-
provoking, well-written, and a clever use of existing datasets. Below are a few comments to improve
clarity and avoid misleading arguments.

We thank the reviewer for their supportive comments.

1. Is there a significant difference between the skew for temperature response for the fungi/oomycetes
grown in the axenic media compared to the other rates measured? It seems like a large portion of your
interpretation is based on this skew being different, but I don’t see it actually stated anywhere. Can
you add P values to Figures 1 and 2?

Please refer to Supplementary Table 2 for statistical analyses of T_{min} , T_{opt} , T_{max} , T_{range} and Skew.
Actually, skew is not key to our arguments regarding Growth in Culture representing the Fundamental
Niche and *in planta* processes representing the restricted Realized Niche – this argument is actually
dependent on the smaller T_{range} for *in planta* processes. The significance of differences for skew was
marginal in the original analysis, and after improvements to the data made on the Reviewers’
suggestions, some of these are now marginally non-significant.

2. The understanding of temperature response needs to be improved/updated. For example, the paper
cited for “Rates increase with temperature to T_{opt} following thermodynamic expectations, followed
by rapid decline in rate as enzymes denature” (Schulte 2015), actually includes text challenging the
viewpoint that enzymes denature following T_{opt} (see Hobbs et al. 2013, ACS chem biology).

We thank the reviewer for bringing this to our attention. Brevity simplified our argument and we have
now included an expanded explanation citing the Macromolecular Rate Theory of Arcus et al. (2016).
(line 69).

3. The discussion of the phylogenetic components could be improved. Can you calculate the
phylogenetic signal for all of the species in the dataset together? It might be that temperature response
is highly constrained based on physiological characteristics of the organism/enzymes.

We would have loved to include such an analysis but a well-resolved, multi-species phylogeny is only
available for genus *Phytophthora* currently. We have noted this in the Methods (lines 466 – 468) that
the phylogeny used relates to *Phytophthora* species. We would welcome further analyses using
phylogenies of other species for which cardinal temperature and/or host species data are available.

4. It would be helpful to include a discussion of the biases of the dataset. For example, more T_{min}
than T_{max} values could correspond to a biased T_{range} value, which is the more important indicator of
the thermal niche breadth. The T_{min} and T_{max} values could additionally be biased because
temperatures measured are chosen by the experimenters. Also, overabundance of particular type of
organism in the analysis could lead to misrepresentation of the results.

More T_{\min} than T_{\max} values, or temperatures chosen by experimenters, would lead to error in the
 dataset rather than bias (because the chosen values could be above or below the true value). Including
 large numbers of species allows us to detect signal despite these errors. We provide information
 regarding how the cardinal temperature data were recorded in the Methods section (lines 322 – 349),
 as well as further information, where necessary, for each row of data in the Togashi dataset. Also,
 please see response to Reviewer 1 (line 90 onwards) concerning error associated with how cardinal
 temperature data were recorded.

We acknowledge that our data are not a random sample of all fungi and oomycetes (we include a
 sentence on this in the Discussion). However, it is difficult to determine what potential biases in the
 results this could have created.

5. I don't understand how you can make claims about not responding to biotic or abiotic niche axes
 when you only test a few components associated with biotic and abiotic niches. Can you de-generalize
 these types of statements in the manuscript or add additional clarification?

In the literature, temperature response is by far the most commonly discussed abiotic niche
 (particularly in Environmental Niche Modelling), and host-range the most commonly discussed biotic
 niche (e.g. food web literature). Hence, we feel justified in using these variables as exemplars of
 abiotic and biotic niche axes. However, we have added qualifiers in the Abstract and Introduction,
 and, we add a sentence in the Discussion to restate (as in the Introduction) that other niche axes exist,
 as described by Hutchinson's n-dimensional hypervolume model.

**Reviewer #3 (Remarks to the Author):**

Chaloner et al has prepared manuscript entitled "Geometry and evolution of the ecological niche in
 plant-associated microbes," which addresses the timely and exciting topic of niche geometry and
 specialization across niche axes in plant pathogens. This is a question I have long wondered about and
 am excited to see investigated. I commend the authors for the strong conceptual and applied
 motivations they put forth for study. Thanks to the authors for a well-motivated and conceptually
 engaging framework!

We thank the reviewer for their very positive response to our work.

I did have some comments and questions about way in which the authors carried out the work:

I know there is significant limitations on the length of the paper. However, there main text does not
 cite the data sources. If the methods are not in the printed paper (which maybe they are based on my
 recollection of other Nature Communications papers), the authors should at a minimum refer the
 readers to the supplementary methods that explains where that data comes from.

The references to cardinal temperature data sources have been added to the main text as follows –

We collated and analysed experimentally-derived temperature responses, specifically the minimum
 (T_{\min}), optimum (T_{opt}) and maximum (T_{\max}) temperatures that comprise the 'cardinal temperatures', of
 various biological processes for 692 plant-associated microbes (628 fungi and 64 oomycetes) cited in
 ref.¹⁵ (Fig. 1). (lines 52 – 55)

T_{opt} and T_{\max} of GC for 101 *Phytophthora* species extracted from ref.²⁰ were used for this analysis.
 (lines 139 – 140)

Does the data collected on the thermal niche of the fungi and oomycetes account for within species
 variation in thermal tolerances?

We averaged data for each species, see Data S2 for number of data points per species in the Togashi
 dataset. There was insufficient information provided in the primary reference (Togashi, 1949) to
 allow unambiguous characterization of within-species variation. For example, data for strains are
 sometimes provided, but the genetic relationships among strains are unknown. This is because all data
 were collected prior to molecular techniques required for this. For example, strains are often cited in
 relation to a particular host i.e. apple strain, or a particular code, i.e. 8L, and not genetic relationship.

We are aware that a small number of studies have found temperature response variation within
 species (e.g. Zhan, J., & McDonald, B. A. (2011). Thermal adaptation in the fungal pathogen
 *Mycosphaerella graminicola*. *Molecular Ecology*, 20(8), 1689–1701), but these are neither sufficient
 in number nor reported in a usable manner, to be included in our analysis.

To understand the quality of these printed estimates of cardinal temperatures would require more
 description of how the data was originally collected. How many data points went into the T_{opt} , T_{max} ,
 and T_{min} of a species?

See Data S2 for number of data points per species, for each biological process, for T_{min} , T_{opt} , T_{max} , in
 the Togashi dataset. To improve the readers understanding of how the data were processed from
 Togashi, (1949), we have included a brief explanation detailing the background of this reference, as
 follow: “In brief, ref.¹⁵ is a compilation of published literature regarding plant pathogen temperature
 relations, published (in print) in 1949. Ref.¹⁵ contains over 300 pages of data from more than 1000
 publications (published in the 19th and 20th century). To our knowledge ref.¹⁵ has not been digitised.
 This publication hence carries a wealth of data poorly accessible to the scientific community, which
 hitherto has not been rigorously interrogated.” (lines 251 – 256).

Are these points from identical lab strains or do these estimates account for within species variation
 among geographically distant strains? Summarizing some of this information about the data early on
 would help readers understand what is being done.

Togashi, (1949) rarely provides information concerning the location the microbial specimen was
 collected i.e. Europe or America. This, along with reasons detailed above, is why we make no
 assessment of within-species variation in cardinal temperature, and average T_{min} , T_{opt} , and T_{max} where
 multiple estimates are provided in Togashi, (1949).

The authors mention that the two databases were used to identify synonymous names for each species
 and were accessed between “1/5/2017 – 18/10/2019” This window seems long as by the time it is
 published it will have been three years. Similarly in the section on host associations later in the
 methods the authors mention that the data on known pathogen/host interactions from the PlantWise
 database (CABI) was accessed more than 6 years ago (28/10/2013). This means some of the
 classifications and host breadths have likely changed, but in my opinion this shouldn’t stand in the
 way of publication as realistically data processing takes time.

Collation and analysis of the data was a lengthy and involved task, hence the duration of the study.
 We thank the reviewer for their understanding in this matter.

I am very concerned about this data collation decision described by the authors as: “If cardinal
 temperature data associated with multiple, nonsynonymous species did not specify which species the
 data referred to, data were recorded under the first species given in ref. 30. For example, in ref. 30 GC
 and DD cardinal temperature data were recorded for *Mycosphaerella tulasnei* (Jancz.) [syn.
 *Cladosporium herbarum* (Link)]. The SFD classified *Mycosphaerella tulasnei* (Jancz.) Lindau (1903)
 as *Mycosphaerella tassiana* (De Not.) Johanson (1884), but *Cladosporium herbarum* (Pers.) Link
 (1816) as *Cladosporium herbarum* (Pers.) Link (1816). However, it was not clear in ref. 30 which
 cardinal temperatures referred specifically to which species. Hence, all data were recorded under
 *Mycosphaerella tassiana*.” Instead of assigning all the data to the first species that happened to be
 listed in ambiguous cases in ref 30, these data should have just been excluded since the authors could
 not determine which species the information was associated with. How many species did this affect?
 What happens to the results when the ambiguous cases are excluded?

We have added substantively to the Methods section on our approach to microbial species
 nomenclature (lines 260 – 311). We have improved our methodology by including additional
 processing steps of species names. We wish to clarify that, where cardinal temperature data associated
 with multiple, nonsynonymous species did not specify which species the data referred to. Rather, we
 chose the first species name provided in ref.¹⁵ because it was the title name given to that record in
 ref.¹⁵. This clarification has been added to the relevant Methods section. We have replicated our main
 figures using the unambiguous subset (Supplementary Fig. 3) showing that our results are unaffected.

In the methods statement “For the Magarey dataset, pathogen names were updated according to the
 SFD or Mycobank [accessed between 1/5/2017 - 18/10/2019] to ensure correct matching to the
 Togashi dataset. No additional processing was performed on the Martin dataset.” After the complex
 set of decisions explained for the Togashi dataset, it leaves the reader wondering why no explanation
 of decisions of t_{max} , t_{min} , t_{opt} was needed for either of these datasets and why looking up names in
 SFD/Mycobank, etc wasn’t necessary for the Martin dataset. A little more explanation would be ideal.

We have clarified by adding the text below. We would also like to highlight here that the Magarey
 and Martin datasets required far less processing than the Togashi dataset, as they are far smaller and
 less complex. “For the Magarey dataset, cardinal temperatures were recorded as point estimates and
 pathogen names were updated according to the SFD or Mycobank [accessed between 1/5/2017 -
 6/3/2020]. (lines 342 – 344). Finally, for each data point in the Martin dataset, where ref.²⁰ recorded
 that the ‘true’ value lies above or below the value provided, the value was used, and where a range
 was provided, the mid-point was used. To ensure maximum matching to *Phytophthora* species
 phylogenies (detailed below), *P. katsurae* was renamed *P. castaneae* in the Martin dataset. We
 assumed that *P. ipomoea* corresponded to *P. ipomoeae*” (lines 345 – 349)

Authors say “All analyses were performed in R 3.2.336. In all analyses the mean of T_{min} , T_{opt} or
 T_{max} for a given biological process, for a given species, is treated as a single datapoint.” Treating
 each species as a single data point makes sense, but how did they deal with species with multiple
 measurements of t_{min} , t_{max} , or t_{opt} ?

We used the mean where multiple measures were available. See Data S2 for number of data points per
 species, for each biological process, for T_{min} , T_{opt} , T_{max} .

Authors wrote in the Analysis of cardinal temperature section and the Niche co-specialisation sections
 that “The Togashi dataset was used for this analysis.” Why only that dataset?

The Magarey dataset was only used for validation (Supplementary Fig. 3), as it only contains data
 regarding the infection process. The Martin dataset was not used here, as it only contains data
 concerning T_{opt} and T_{max} , hence no estimates of T_{range} are possible. The Togashi dataset contains more
 than 8000 data points so we focussed on this.

Also later in the paragraph on that analysis the authors say “In some cases, for particular species-
 biological process combinations, mean T_{opt} was estimated as greater than mean T_{max} or lower than
 mean T_{min} . This is because data were extracted from multiple sources.” I’m confused about what is
 meant by multiple sources since only Togashi is used for this analysis. Similarly, later still in the
 section the authors say “Species with at least one T_{opt} , T_{range} or skew estimate were included in
 analyses involving T_{opt} , T_{range} and skew, respectively.” What is meant by “at least one”? Does this
 mean that in some cases species have multiple measurements of T_{opt} or of T_{range} ? I suspect a lot of
 this confusion would be circumvented with more explanation of the dataset.

We apologise for any confusion. Where we stated “This is because data were extracted from multiple
 sources”, we were referring to multiple records within Togashi, (1949). It is correct that only a single
 source was used for this analysis, but this single source may cite cardinal temperature data for a
 particular species from multiple, independent sources. We have clarified this with the two sentences
 below:

“Where multiple, independent cardinal temperature estimates were cited for the same species and
 biological process in ref.¹⁵, the mean was taken.” (lines 379 – 381)

“In some cases, for particular species-biological process combinations, mean T_{opt} was estimated as
 greater than mean T_{max} or lower than mean T_{min} . This is because data from multiple, independent
 sources were provided within ref.¹⁵. In such cases, nonsensical values (i.e. skew < 0 or > 1) were
 removed for these species-biological process combinations.” (lines 396 – 399)

What does the '(224)' in the following sentence mean? "235 (224) pathogens of hosts included in the
processed host phylogeny were identified with at least one Trange (Trange(50%)) estimate, for at
least one biological process, in the Togashi dataset (Supplementary Table 7)."

We apologise for any confusion. Above, we were referring to the number of pathogens included in
analyses concerning Trange and Trange50%. We have clarified this by summarising this information
in Supplementary Table 6. The number of pathogens in this table are greater than that detailed above,
because our improved methods for identifying pathogens in the PlantWise database increased the
sample of pathogens used in this analysis.

In the supplementary figure1 on the "Analysis of spatial correlation on phylogenetic signals
calculated for Phytophthora species cardinal temperatures", the caption mentions that the p-values for
the mantel correlations are all less than <0.05 . Then the authors say that "The Mantel correlations are
near zero hence can ignore the influence of spatial effects on our analysis of cardinal temperature
phylogenetic signal". I may be misunderstanding the meaning of this statistical test, but this statement
that we can ignore spatial effects seems incongruent with the significant p-values.

The very large sample size in our dataset (1.37 million datapoints) means that even very tiny effects
(Mantel correlations around $-0.01 - 0.04$) become statistically significant. However, the correlation is
so weak that spatial effects can be ignored as explanations for the phylogenetic signal we have
detected.

Supplementary Figure 3 is especially helpful.

Thank you.

Reviewers' Comments:

Reviewer #1:

Remarks to the Author:

I thank the authors for their responses to a number of the questions raised by myself and by other reviewers. Overall, I think this is a really interesting contribution.

In some cases there have been changes that have clarified confusing points or appropriately addressed questions.

However, there are a two concerns that were essentially dismissed without significantly addressing either in responses or in the manuscript. I'm not sure that they are "fatal" problems, but lack of addressing them certainly weakens a manuscript that could be much more solid.

For instance, Reviewer 1 asked if there has been any historical conflation of specialist/generalist between the abiotic conditions and host range, and if not why one might expect that this would be conflated. The answer given is that the "literature has never differentiated specialism on biotic and abiotic niche axes." and this is the first to address the question. There is still no indication why this would be interesting, only that it is the first. Just because it has not been addressed before does not make it an interesting question. Similarly, the authors say the reason that the claim that the realized niche is smaller than the fundamental niche is important here because it has not been tested is not compelling; this is so by definition, and so it is not surprising that it has not been tested. I think these are a pretty important issues to address because otherwise the paper is solving a problem that no one previously recognized and that the authors do not provide a clear explanation about why it might be a problem.

Each of the reviewers had questions about the Cardinal temperatures data sets, including asking for an "assessment of the impact of how data were handled 91 when the cardinal temperatures were "above #C" or "below #C", "biases of the dataset", and "What happens to the results when the ambiguous cases are excluded?". In the last case, the authors actually did the analysis eliminated the ambiguous cases to show what the impacts was (Suppl Fig. 3). However, in the first two, they provide some discussion, but what would be much better is actually assessing what the impact of the assumptions made would be on the outcome of the analyses. Since these temperature values at the extremes are critical to the analyses, an empirical assessment of what the impacts of assumptions about those values would be on the conclusions of the analyses is important.

Reviewer #2:

Remarks to the Author:

The authors did a great job responding to the reviewer comments and editing their manuscript appropriately. Prior concerns about interpretation of results were adequately addressed and explained in the new version of the manuscript.

Reviewer #3:

Remarks to the Author:

I am happy with the author's responses to my reviews and look forward to seeing this work published. This is a really interesting topic and well-motivated paper.

**Geometry and evolution of the ecological niche in plant-associated microbes**

Chaloner TM, Gurr SG & Bebbler DP

Response to reviewers' comments

**Reviewer 1**

I thank the authors for their responses to a number of the questions raised by myself and by other
reviewers. Overall, I think this is a really interesting contribution.

In some cases there have been changes that have clarified confusing points or appropriately addressed
questions.

However, there are a two concerns that were essentially dismissed without significantly addressing
either in responses or in the manuscript. I'm not sure that they are "fatal" problems, but lack of
addressing them certainly weakens a manuscript that could be much more solid.

For instance, Reviewer 1 asked if there has been any historical conflation of specialist/generalist
between the abiotic conditions and host range, and if not why one might expect that this would be
conflated. The answer given is that the "literature has never differentiated specialism on biotic and
abiotic niche axes." and this is the first to address the question. There is still no indication why this
would be interesting, only that it is the first. Just because it has not been addressed before does not
make it an interesting question.

*The concept of the niche is central to the science of ecology – since the foundational work of*
*Hutchinson, the niche has been conceived as a shape in n-dimensional niche space. Given that*
*Hutchinson's model is explicitly geometric, we feel that understanding the shape of the niche is a key*
*scientific goal. In addition, the lack of correlation between niche breadth on abiotic and biotic axes*
*has profound consequences for our understanding of the evolution of specialization, and of the*
*interactions between plant pathogens and their hosts. We have included statements on these*
*arguments in our Discussion and feel that further exposition would not add substantively to the*
*manuscript.*

Similarly, the authors say the reason that the claim that the realized niche is smaller than the
fundamental niche is important here because it has not been tested is not compelling; this is so by
definition, and so it is not surprising that it has not been tested. I think these are a pretty important
issues to address because otherwise the paper is solving a problem that no one previously recognized
and that the authors do not provide a clear explanation about why it might be a problem.

*We do not claim that the difference between the realized and fundamental niches has not been tested,*
*in fact we cite a recent example from the literature on reptiles and amphibians. We claim that ours is*
*the first example for microbes in general and plant pathogens in particular, and the first example*
*derived from direct physiological measurements rather than being inferred from species distributions.*
*Further, it is a hypothesis that the realized niche is narrower than the fundamental niche due to biotic*
*interactions, not a definition. The degree to which the fundamental niche is restricted by biotic*
*interactions is an important question in ecology as it determines the observed distribution of species,*
*and the degree to which species' ecology is shaped by interactions with other species.*

Each of the reviewers had questions about the Cardinal temperatures data sets, including asking for an
"assessment of the impact of how data were handled 91 when the cardinal temperatures were "above
#C" or "below #C", "biases of the dataset", and "What happens to the results when the ambiguous
cases are excluded?". In the last case, the authors actually did the analysis eliminated the ambiguous
cases to show what the impacts was (Suppl Fig. 3). However, in the first two, they provide some
discussion, but what would be much better is actually assessing what the impact of the assumptions
made would be on the outcome of the analyses. Since these temperature values at the extremes are

critical to the analyses, an empirical assessment of what the impacts of assumptions about those
values would be on the conclusions of the analyses is important.

We believe that uncertainties in cardinal temperature data could add error to our dataset, not bias.
However, we can appreciate the reviewer's concerns, and we acknowledge that we did not formally
address this issue with further analyses in our previous submission. We hope we have adequately
addressed this issue by adding the following to the Methods of our study:

**Influence of uncertainty in reported cardinal temperatures:**

Cardinal temperature data in the Togashi and Martin datasets may contain uncertainties due to
reported values varying from their true values (i.e. $T_{\max} < 32$). We investigated whether these potential
uncertainties could have affected our conclusions concerning fundamental vs. realised niche
geometry, niche cospecialisation, and cardinal temperature phylogenetic signal. It was beyond the
scope of this study to establish how cardinal temperatures were determined for each record reported
ref.15. Further, ref.20 does not provide information or references as to how reported T_{opt} and T_{\max}
were determined. To overcome this, we assumed that in all cases cardinal temperature was
investigated experimentally at 5 °C increments, and that the minimum and maximum temperature
treatments spanned T_{range} . This implies that cardinal temperature estimates do not deviate from their
true values by more than ± 2.5 °C. Hence, for both the Togashi and Martin datasets, 2.5 °C was added
or subtracted from cardinal temperature data reported as being above or below their true
value, respectively. For example, GC data reported as > 25 °C was modified to 27.5 °C. Where data in
the Togashi dataset were extracted from prose, data were modified in this way only if we could
determine the direction of the error with confidence. In all cases, these uncertainties have little effect
on our results and did not affect any of our key conclusions (Supplementary Tables 7, 10, 13). This
suggests that uncertainties are randomly distributed and do not affect the results presented here.

**Reviewer 2**

The authors did a great job responding to the reviewer comments and editing their manuscript
appropriately. Prior concerns about interpretation of results were adequately addressed and explained
in the new version of the manuscript.

Thank you

**Reviewer 3**

I am happy with the author's responses to my reviews and look forward to seeing this work published.
This is a really interesting topic and well-motivated paper.

Thank you.